# Sequence-oriented sensitive analysis for PM2.5 exposure and risk assessment using interactive process mining

**Eduardo Illueca Fernández**[1,3]*, **Carlos Fernández Llatas**[3,4], **Antonio Jesús Jara Valera**[2], **Jesualdo Tomás Fernández Breis**[1], **Fernando Seoane Martinez**[3,5,6,7]

**1** Department of Informatics and Systems, University of Murcia, Murcia, Spain, **2** Research and Development, HOP Ubiquitous S.L. (Libelium Murcia), Ceutí, Spain, **3** Department of Clinical Science, Intervention and Technology, Karolinska Institutet, Stockholm, Sweden, **4** ITACA-SABIEN, Polytechnic University of Valencia, Valencia, Spain, **5** Department of Medical Technology, Karolinska University Hospital, Stockholm, Sweden, **6** Department of Textile Technology, University of Borås, Borås, Sweden, **7** Department of Clinical Physiology, Karolinska University Hospital, Stockholm, Sweden

* eduardo.illueca@um.es

**Data Availability Statement:** All data and code files are available from the Zenodo repository (http://doi.org/10.5281/zenodo.8079155).

## Abstract

The World Health Organization has estimated that air pollution will be one of the most significant challenges related to the environment in the following years, and air quality monitoring and climate change mitigation actions have been promoted due to the Paris Agreement because of their impact on mortality risk. Thus, generating a methodology that supports experts in making decisions based on exposure data, identifying exposure-related activities, and proposing mitigation scenarios is essential. In this context, the emergence of Interactive Process Mining—a discipline that has progressed in the last years in healthcare—could help to develop a methodology based on human knowledge. For this reason, we propose a new methodology for a sequence-oriented sensitive analysis to identify the best activities and parameters to offer a mitigation policy. This methodology is innovative in the following points: i) we present in this paper the first application of Interactive Process Mining pollution personal exposure mitigation; ii) our solution reduces the computation cost and time of the traditional sensitive analysis; iii) the methodology is human-oriented in the sense that the process should be done with the environmental expert; and iv) our solution has been tested with synthetic data to explore the viability before the move to physical exposure measurements, taking the city of Valencia as the use case, and overcoming the difficulty of performing exposure measurements. This dataset has been generated with a model that considers the city of Valencia's demographic and epidemiological statistics. We have demonstrated that the assessments done using sequence-oriented sensitive analysis can identify target activities. The proposed scenarios can improve the initial KPIs—in the best scenario; we reduce the population exposure by 18% and the relative risk by 12%. Consequently, our proposal could be used with real data in future steps, becoming an innovative point for air pollution mitigation and environmental improvement.

**Funding:** The author EIF has received funded from Fundacion Séneca (https://fseneca.es/), grant number 21300/FPI/19 The authors have received funded from EIT Health (https://eithealth.eu/), grant number 220649 The funders had no role in study design, data collection and analysis, decision to publish, or preparation of the manuscript.

**Competing interests:** The authors have declared that no competing interests exist.

## Introduction

Air pollution is the first health risk significantly affecting morbidity and mortality [1]. The particular importance is the effect of aerosol and particle matter (PM). Despite the considerable improvement in European air quality over the past decades, in 2019, approximately 74% of the urban population was still exposed to PM2.5 concentrations exceeding the World Health Organization (WHO) air quality guidelines for health and 70% of official air quality monitoring points in the European Union report values above these limits. These values are expected to increase due to the movement of the population to cities [2]. For this reason, it is estimated that particle matter belongs to the five highest mortality risk factors worldwide, causing 4.2 million premature deaths [3]. In addition, other studies show that more than half of the mortality burden can be attributed to exposure to suspended particles of aerodynamic diameter less than 2.5 $\mu g/m^3$ (PM2.5) [4] and these particles can trigger adult vascular disease events [5].

In this context of increasing morbidity, it is a key point to estimate the cost of these risks. The health impact of a pollutant—such as PM2.5—is related to the population's exposure to this pollutant. The exposure of a single individual could be easily computed as the sum of the product of time spent by a person in different environments and the time-averaged pollutant concentrations occurring in those locations [6]. Thus, population exposure is an aggregation of the exposure of all individuals who belong to those populations, and it is necessary to understand the health risk in specific population groups. To quantify this effect, integrated exposure-response functions estimate the relative risk (RR) associated with PM2.5. These functions are models that relate the exposure to the relative risk [7]. Thus, it is possible to deduce that there are two main indicators to evaluate the health impact of air pollution: i) the population exposure, which is homogeneous for all citizens and ii) the relative risk that could take into account other epidemiological factors and other health risks present in the population.

To calculate these indicators, it is necessary to define a model that considers two inputs: (1) the geospatially distributed PM2.5 concentration—or the relevant pollutant—and a geolocated list of the activities performed by each member of the population; and (2) the duration of each of these activities. This approach is employed by the EXPLUME model, which uses as inputs the simulations performed by the CHIMERE chemical transport model to simulate the geospatial distribution of PM2.5 concentration, and the list and duration of activities are based on data from the French Institute of Statistics [8] in the city of Paris. The main limitation of this study is that it does not integrate each individual's activities. In other words, a list of activities is generated only for the aggregate population, and the activity's effect on the overall scenario is not considered. In our view, this limits the model considerably and leads to a loss of information because it is not possible to track how pollution affects each individual, and it is not possible to compute individual KPIs. Thus, sequential data and sequence-oriented analysis are critical to overcoming these limitations.

Therefore, applying interactive process mining to the traces describing the activities can help to exploit this information more comprehensively [9]. We hypothesise that the application of interactive process mining will allow us to perform a sequence-oriented sensitivity analysis and identify the best scenarios for reducing the population's exposure to PM2.5 and the associated risk more quickly and accurately. To test this hypothesis, a synthetic dataset will be generated with a methodology similar to that proposed by EXPLUME [8], using data from the Spanish Institute of Statistics (https://www.ine.es/) for the city of Valencia (Spain) and interactive process mining will be applied to select the best scenarios, using personal exposure and relative risk as KPIs. These indicators will then be recalculated for the proposed strategies, quantifying the improvements. The application of interactive process mining for exposures assessment is completely new in state of the art and has the following advantages: i) it allows to analyse and

quantify the exposure of each individual in the population, as well as the activities they perform; ii) it generates knowledge in an understandable way for public authorities and citizens; and iii) it allows to exploit the new infrastructures related to Smart Cities, where information about citizens' activities can be obtained through mobile applications and IoT devices [10].

The rest of the paper is structured as follows. First, the *State of the Art* section sets the theoretical background and the work performed in air quality modelling and exposure assessment, as well as the approaches to estimate relative risk. Then, the *Materials and Methods* section describes the following points: i) the methodology used for modelling and the strategy for the generate the population and sequence data; ii) the interactive process mining workflow and iii) the KPIs computation. From these simulations and computations, a set of *Results* is obtained that provides the effectiveness of each scenario. Then, these results and the feasibility of the proposed methodology are analysed in the *Discussion* section and compared with state of the art, obtaining the next steps for the implementation in a real pilot. Finally, the main conclusions and future challenges are presented in the *Conclusions*.

## State of the art

In the last years, a large amount of research has been performed to assess the health impact of particle matter and its relationship with the exposure population. These impacts can classify into long-term effects and short-term effects. Particulate matter pollution originates from a wide range of anthropogenic and natural sources, and its characteristics can vary in size distributions, chemical composition, and other properties. The resulting health outcomes also vary substantially, depending on an individual's target physiological system or organ [11]. For this reason, the health outcomes are diverse and not well-known. Some cohort studies reported an excess risk for all-cause and cardiovascular mortality due to long-term exposure to PM2.5 [12] and the exposure to particles results on immediate effects on respiratory admissions [13].

To quantify this effect on public health, it is necessary to determine the exposure of the population to the pollutant, for which two inputs are needed: i) the personal activities performed during an interval of time, and ii) the air quality in the environment where the activity is performed. Regarding the first point, the main approach in state of the art is using individual sensors to monitor the activities of the citizens and the output concentration of the pollutants [6]. However, collecting this activity data could be complex and should require the consent of the participants in the experiment. One alternative consists of asking participants to finish a diary questionnaire that includes time, activities, and location. This approach has been applied in the school context [14]. On the other hand, a more general strategy is to model this activity data using different strategies [8]. For instance, activity patterns—which specify how long a person stays in each microenvironment—were derived from the advancement of the Multinational Time Use Study MTUS, which combines more than a million diary days from over 70 randomly sampled national-scale surveys into a single standardised format. MTUS allows researchers to analyse time spent by people in various sorts of work and leisure activities over the last 55 years and across 30 countries [15].

On the other hand, these activities must be cross-checked with particle matter values. For this, a concentration value is needed for each of the geolocations of the activities. In that sense, developing air quality models has facilitated the generation of spatially continuous data. The most robust eulerian model in state of the art regarding air quality and particle matter is CHIMERE, which allows for obtaining a concentration value for each of the cells of a grid [16]. However, in the context of exposure calculations, a high resolution is required that classical models do not have. Two strategies are generally used to improve the resolution of air quality data: i) the use of Street Canyons models, which simplifies some of the equations [17], and ii)

the downscaling approaches based on linear regression models [18]. These models also allow the estimation of particle trajectories, especially when combined with Lagrangian models such as HYSPLIT [19]. It should be noted that all these models have been validated in real contexts, such as the study carried out in the city of Milan [20].

However, the exposure is not enough to assess the health risk. For this reason, WHO and Europe recommended 2015 a set of linear concentration–response functions for the main air pollutants and related health outcomes [21]. These functions are currently widely used for health assessments, e.g. on a European scale by EEA. In contrast, it is currently widely debated what the optimal shape of the concentration–response functions is and whether there should be a threshold or lower limit and the use of the linear functions [11]. In consequence, and beyond these exposure-risk functions, more complex models have been developed, such as the Global Exposure Mortality Model (GEMM) functions. For PM 2.5, the non-accidental mortality generally follows a supralinear association at lower concentrations and a near-linear association at higher concentrations, showing that health impacts related to PM 2.5 exposure have been underestimated at both the global and regional scales [22]. This model has been used recently to estimate cardiovascular mortality, obtaining a value towards 790 000 premature deaths [23].

Nevertheless, the methods presented in the state of the art do not consider a fundamental factor in an exposure study: time. Each activity registered in this kind of dataset is associated with a timestamp and a duration. Therefore, the application of new technologies in the data science field can help deal with new challenges related to timestamp management. In this context, process mining represents a collection of tools, methods, techniques, algorithms, etc., that allows achieving a better understanding of the execution of a process by means of analyzing the operational execution data that is generated during the execution of the process [24]. This approach fits with the activity sequence analysis, and this study is the first contribution—to hour knowledge—of process mining for individual exposure assessment in state of the art.

## Materials and methods

### Fundamentals and basic concepts

The atmosphere, whether urban or remote areas, contains many aerosol particles suspended. From a physical point of view, particle matter (PM) is a mixture of solid particles and liquid droplets found in the air. Some particles, such as dust, dirt, soot, or smoke, are large or dark enough to be seen with the naked eye. Others are so small they can only be detected using an electron microscope. This wide size range can be appreciated by considering that the mass for one 10 $\mu m$ is equivalent to the mass of one billion 10 nm particles. Thus, working with each particle as a single entity is difficult. It is necessary to work with particle populations characterised by a cumulative size distribution, defined as the particles that are smaller than or equal to this size range. From this cumulative distribution, the concepts PM10, PM2.5, and PM1 arise, which are particles smaller than or equal to 10 $\mu m$, 2.5 $\mu m$, and 1 $\mu m$, respectively [25].

This work focuses on the PM2.5 population—also called the PM2.5 fraction—because of the impact on health. A percentage of a particle population can pass through the alveolar barrier in the lungs and reach the blood torrent. This percentage of particles is the respirable fraction responsible for adverse health effects and premature deaths. The size of these respirable particles depends on their chemical composition and the corporal weight of the individual. The percentage of particles smaller than PM2.5 that can pass the alveolar barrier is close to 100%, showing the need to focus mitigation policies on this dangerous pollutant [26].

For this elaborate effective strategy, measuring the impact of a concentration of PM2.5 in the air is necessary. This is done by computing KPIs. The exposure is the closest to

concentration, and it has been previously defined as the number of particles a citizen is exposed to in an interval of time. This amount of pollutant affects the individual and can be modelled as the effect of other drugs or toxic. In this sense, concentration-response functions act as a model that measures this effect in a concrete outcome—generally, this is a pathology. There are a lot of concentration-response functions that link PM2.5 with several health issues, but this work will focus on the effect of PM2.5 on premature mortality [21].

## Synthetic data generation

The final objective is to generate a dataset with the activities performed by the citizens of a population throughout a working day and relate to each activity the individual exposure and the mortality relative risk. A model has been implemented based on the workflow defined in Fig 1 to generate this dataset, which is detailed in this section. The first step in data generation is the domain definition. In other words, this process involves assigning the perimeter of the use case, the spatial resolution and the locations of the buildings, workplaces, schools, etc. and their properties. This definition is done using the geoJSON standard [27], allowing the addition of georeferenced polygons and points. According to this format, two files are generated: i)

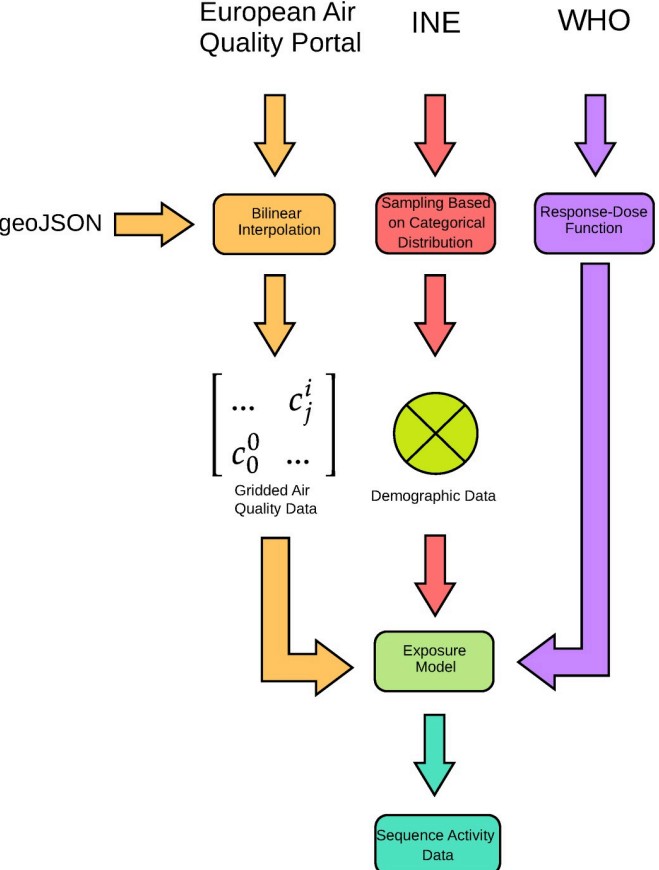

**Fig 1. Synthetic data generation workflow.** First, collected data from different sources is mapped in the domain geoJSON, generating gridded air quality data—after applying a dispersion model—and demographic data. This is combined with the dose-response functions from WHO to generate sequence activity data related with personal exposure and mortality relative risk.

the first one is the gridded domain divided into squared cells—with a resolution of 5 x 5 km—and ii) a file with the coordinates of the locations, buildings and residences where citizens could live. In addition to the georeferenced data, the (geoJSON) format allows adding other attributes to the polygons and points. For the cells, the most important attribute is the cell identifier. For the locations, the identifier and important metadata are generated as the location type and the I/O ratios. This last parameter is required for the personal exposure computation, so it is necessary to define these I/O ratios accurately. For this reason, we use the I/O ratios proposed in this paper for several buildings and activities [28].

Since the analysis will be performed with 2018 data, the air quality data is obtained from the historical data obtained from the air quality stations. This information is in open data format and accessible through the European Air Quality Portal API. The data for the year 2018 have been downloaded for the stations ES1181A, ES1911A, ES1239A, ES1926A, ES1884A, ES1619A and ES1912A. The raw data provides only air quality measurements for seven locations. Still, gridded data is required to compute the individual exposure—a value for each cell defined in the domain. To overcome this problem, a bilinear interpolation model (Eqs 1 and 2) is used to fill the cells with PM2.5 concentration values, using the following formula, where $c_i$ is the PM2.5 concentration in the cell $i$; $c_j$ is the PM2.5 concentration in the station $j$; $w_j$ the weight of the station $j$; n is the number of stations—in this case, n = 7; $d_{ij}$ is the distance between the cell $i$ and the station $j$; and $d_{max}$ is the maximum computed distance.

$$c_i = \sum_{j=1}^{n} c_j * w_j \tag{1}$$

$$w_j = \frac{1}{d(1 - d_{max})} \tag{2}$$

The simulated sample population is composed of 15795 individuals, around 0.2% of the total population of Valencia. The reason to take this value is that the sample is not large for computational purposes but big enough to keep all the information that characterises the population. A unique identifier is assigned to each citizen, and information about age, sex, habits, and pathologies is generated for each one. In addition, each citizen is assigned to one of the residences—locations where a citizen could habit—defined in the geoJSON file and to one workplace. This job could be in the industry, services, agriculture, school, university, or another residence. Last, information about the preferred mean of transport—metro, car or by foot—are added as well as information and location of free-time activities. The statistics to generate this information are taken from the INE (Spanish Institute of Statistics).

The next step is the computation of the sequence activities. For each citizen, the sequence of daily activities has been modelled. This step is critical because it is necessary to know the activities' duration to define the timestamp. In this context, two actions are prescribed: static and movements. For the first one, the computation is not complicated if we assume that all the people work 8 hours per day (6 hours for school and university) and other constant time for static activities. The durations are modified by adding some Gaussian noise to increase the variability. However, for movements, this computation is more complicated. It is necessary to calculate the distance between the start and end points and the time of the movements assuming that the velocity is 5.7 km/h by foot and 40 km/h by car or metro. To assess this sequence of activities, we propose a methodology similar to the one proposed by the EXPLUME model [8].

The exposure of a single individual is computed using as input the gridded air quality data and the activity sequences [6]. The approach proposed suggests calculating the exposure of the citizen $i$ ($E_i$) as the summation of the product of the concentration of PM2.5 that the citizen is

exposed in each daily activity and the time spent in each activity, according to Eq 3, where $c_j$ is the average concentration at the location of the activity $j$, $t_j$ is the activity time $j$ and $a$ is the number of activities realised during a day.

$$E_i = \sum_{j=1}^{a} c_j * t_j \tag{3}$$

However, there are special cases where this formula is not entirely accurate for calculating individual exposure. The most important—because of its relevance during the COVID-19 pandemic—is the use of face masks. In this case, one study [29] proposes the approximation proposed in Eq 4, which assumes a pre-correction to the concentration values before applying Eq 3. In Eq 4, $c$ is the ambient concentration, $c_{mask}$ is the corrected concentration, $PF$ is the mask protection factor, and $f_{day}$ is the fraction of the day used for sampling.

$$c_{mask} = \left(\frac{c}{PF}f_{day}\right) + (c(1 - f_{day})) \tag{4}$$

The relative risk being minimised is the mortality relative risk due to PM2.5 exposure. In other words, we search to quantify the risk of death in the population related to the levels of observed PM2.5. This relative risk is composed of several factors. The first one is the relative risk due to only the exposure to pollutants, and the other components are the risk factors or other diseases that could increase the effect of the particles. The final relative risk is the combination of all these risks.

The relative risk assessment is performed using the AirQ+ software developed by the WHO. The program calculates the magnitude of several health effects associated with exposure to the most relevant air pollutants in a given population. AirQ+ has been validated recently in a Campaign in Tarragona [30] to measure long-term exposure to ambient particulate matter PM2.5. In concrete, this tool has been used to assess the exposure-response function for PM2.5 in Valencia. The following methodology consists in obtaining the historical PM2.5 data for the 2018 year in the seven air quality stations and computing the historical data for the mean of all the stations. These data are used as input for the AirQ+. Once the exposure-response function is obtained, the model is used to compute the relative risk (RR) related to the exposition and the activity. In addition, this mortality relative risk is modified by the factor risks of the individual—diseases, smoking, etc.—by combining all the relative risk that affects the individual. The epidemiology RR is calculated from the death rate of different conditions obtained from the Global Burden of Disease database [31].

## Sequence-oriented sensitive analysis

The main goal of this work is to perform a sensitivity analysis to reduce the exposure and the health outcomes by taking action on the activities. Sequence analysis is essential for our proposed optimisation because this allows for avoiding population metrics that could be biased by the sampling process. Thus, it allows for performing analyses for different population sectors —for instance, analysing only the persons affected by cardiac issues—or even tracking and optimising the activities of a single individual [14]. In addition, the classical sensitive analysis should be performed by doing different simulations among the inputs to find the inputs that minimise a set of KPIs. The computational cost of this sensitive analysis could be high due to the number of input parameters to consider. Thus, we propose a new methodology to assess a sequence-oriented sensitive analysis using interactive process mining (IPM) in this paper.

Process mining is a methodology that uses timestamp information to create human-understandable views that explain a sequence of processes (log). This approach's principal advantage

is avoiding the black box effect in some models in classical machine learning and data science [24]. IPM results from applying Interactive Pattern Recognition methodologies to Process Mining technologies [32]. Interactivity is an essential feature of our semi-automated process discovery, which uses information from the event log and user expertise. That is one of the differential features of our approach. As a result, the sequence-oriented sensitive analysis is composed of the following steps: i) computation of the initial KPIs in the base scenario, ii) ANOVA analysis to identify differences between activities and which activities are related to a high exposure and a high-risk, iii) generation of the process map and significance analysis, iv) proposition of new scenarios and v) computation of the KPIs for the new scenarios and comparison with the initial KPIs.

Interactive process mining is a methodology that requires an iterative human validation by domain experts through the so-called data rodeo. In this way, the domain experts can i) iteratively define a process indicator according to the sensitive analysis goals, ii) analyse and validate the process indicator and iii) be trained in using the process indicator. This process is oriented by an interactive process indicator (IPI)—in this case, IPI can be personal exposure or mortality relative risk. Once an IPI is defined, the domain expert is ready to analyse the data using an interactive process mining tool. This process is repeated until reaching the target scenarios, which should be validated by computing the KPIs again and checking for achieving the optimisation goal, in this case, reducing personal exposure and relative risk. In this work, the involved stakeholders are Libelium as domain experts—an IoT company that provides sustainability impact assessment solutions in smart cities—and data scientists from Karolinska Institutet, ITACA-Sabien, and the University of Murcia. The interactive process mining workflow is summarised in Fig 2.

The interactive process mining workflow is semi-automatic and can be summarised in the following steps. First, the CSV ingestion into the PMApp is done. In this step, it is possible to define filters to remove observations that are not correct; next, the CSV file is transformed into an ECSV file, the proper format for the Process Discovery algorithm. In this work, we use the PALIA algorithm, implemented by the Institute of Information and Communication Technologies (ITACA) of the Universidad Politecnica de Valencia, Valencia, Spain. This algorithm is the most appropriate one for our goals, because it is based on activity-based possess mining and produces explainable process maps [33]. In addition, it performs better, in terms of efficacy, than other process mining algorithms such as heuristic miner [34] or genetic process

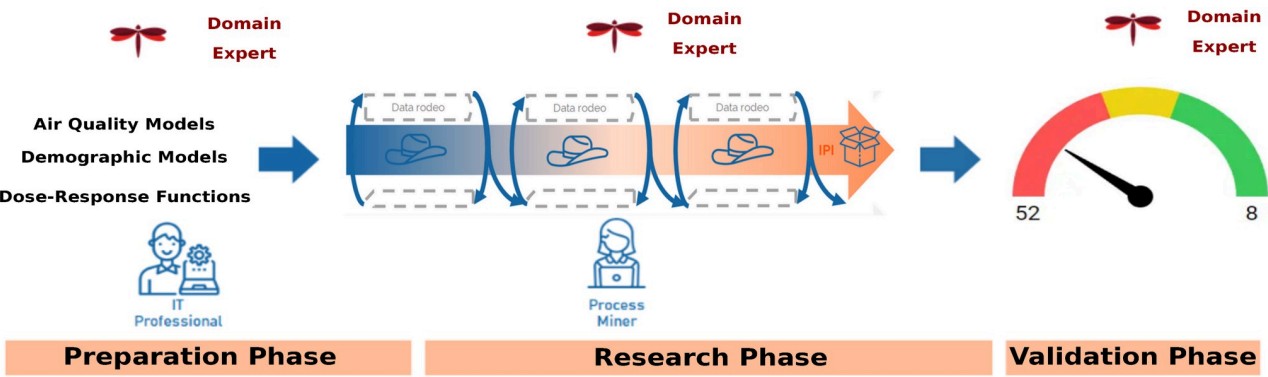

**Fig 2. The first step of the IP is the preparation phase, where the simulations are performed; then, the generated data is processed through interactive process mining in constant discussion with domains expert, and the proposed scenarios are validated with domain experts once the KPIs are calculated again.**

mining [35]. Once the process map has been obtained, the transition probabilities and other relevant characteristics, such as the number of traces, are calculated. The last step is semi-automatic and should be performed according to expert guidelines. It consists of the computation of P-values according to the statics computed before and defining a new scenario.

PMApp is a tool for generating custom PM dashboards to visualise process data coming from health organisations and producing advanced process views provided by the IPIs to empower the analysis made by the health stakeholders. PMApp also enables the creation of interactive dashboards that respond to the selection of arrows and nodes by capturing GUI events. It also allows the user to create custom forms and algorithms for discovery, filters, enhancement maps, etc. In PMApp, it is possible to render maps that enhance the discovered model using colour gradients. With this feature, it is possible to generate specific maps highlighting situations that depend on a customised formulation and are represented by nodes [36]. PMApp implements the PALIA algorithm, which results in the traceability of all learning processes, so each activity is continuously associated with single events [37].

### KPIs computation

The last part of the proposed methodology is the quantification of the improvement implied by the proposed scenarios based on numerical KPIs. The following metrics are proposed for this work: i) 24 hours population exposure, ii) Population Relative Risk, iii) Percentage of Risky Activities and iv) Time Spent in risky activities. These quantifiers are not fixed and can be reformulated depending on the context, but their definition is recommended to be agreed upon between the data scientist and the urban health and environment specialist.

## Results

### Dataset generation and initial KPIs

The model output implies the generation of two output datasets related to the input parameters stored in CSV. The first is the population dataset, which describes all the citizens in the domain. This information includes sex, age, epidemiological data, location of the residence place and the workplace. This dataset comprises 15795 individuals—close to 2% of Valencia's population. On the other hand, the sequences dataset keeps the information of all the movements and activities of a labour day—using as concentration input the historical data of 2018-03-01. Over this sequence dataset, the previously described KPIs are calculated, obtaining a 24-hour Population Exposure of 118 $\mu g *$ h/$m^3$. In addition, the other indicators show a Population Relative Risk equal to 1.90; a percentage of risky activities of 56% and a time spent in hazardous activities of around 3.50 hours. These results are plotted in Fig 3.

### ANOVA analysis

Once the CSV file is generated, with the log info about the sequence data, it is used to perform the ANOVA analysis. For the exposure, we obtain a $F_{value}$ = 14846 that is associated with a $P_{value}$ = 0. If we compare the means in a Tukey posthoc analysis [38], it is observed that the activities with high individual exposure are agriculture and free-time activities. In addition, other work activities like industry, services, or school—that suppose a huge amount of time—are related to high exposure. These values are detailed in Fig 4, where the ANOVA table is included, as well as a heatmap showing the exposure mean differences between activities.

Regarding the population relative risk, the ANOVA analysis computes $F_{value}$ = 739 and $P_{value}$ = 0, showing that there are significant differences between the risk of activities, so it is possible to define risky and safe activities. The high-risk activities are agriculture, leisure, and

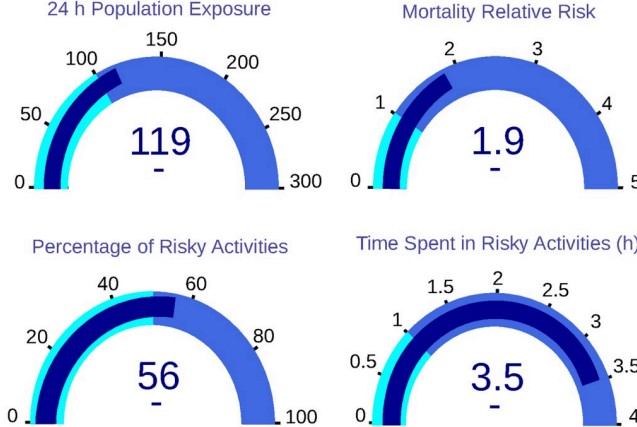

**Fig 3. Dashboard with base KPIs of the sample population.** In concrete, 24 h population exposure; mortality relative risk; percentage of risky activities and time spent in risky activities.

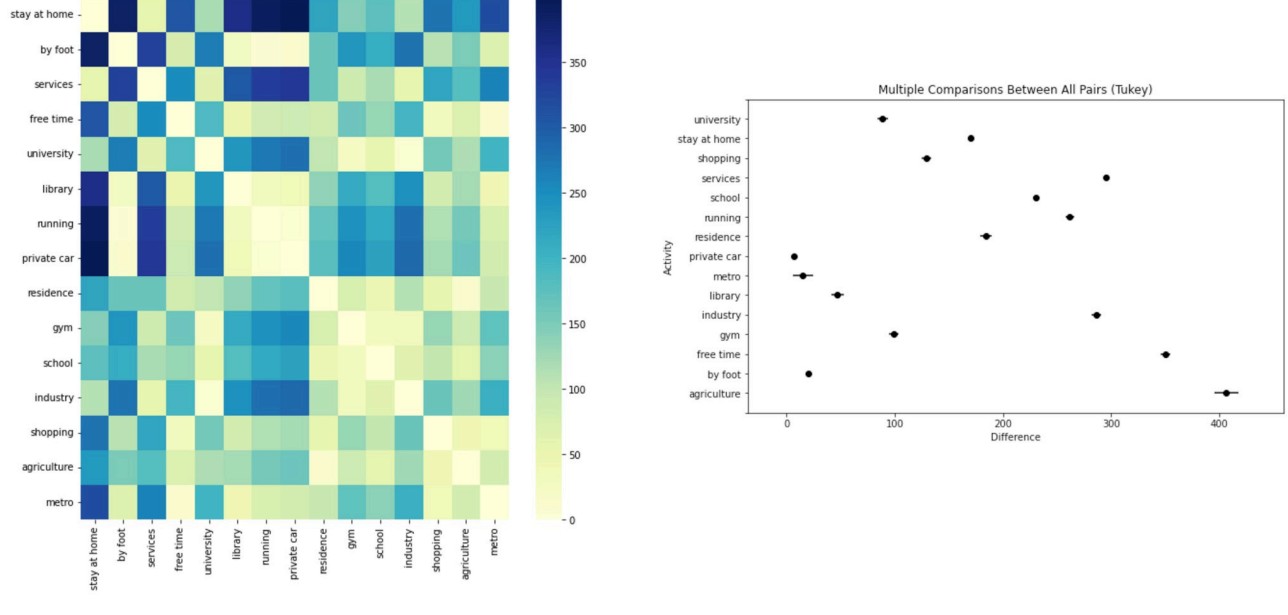

| | SS | DF | F-VALUE | P-VALUE |
|---|---|---|---|---|
| Activity | 1.06e+9 | 14.0 | 14846 | 0.0 |
| Residual | 5.17e+8 | 101170 | - | - |

**Fig 4. ANOVA analysis for the exposure among activities.** The ANOVA table is shown in addition to the heatmap with the exposure difference between activities and the plot of the Tukey test.

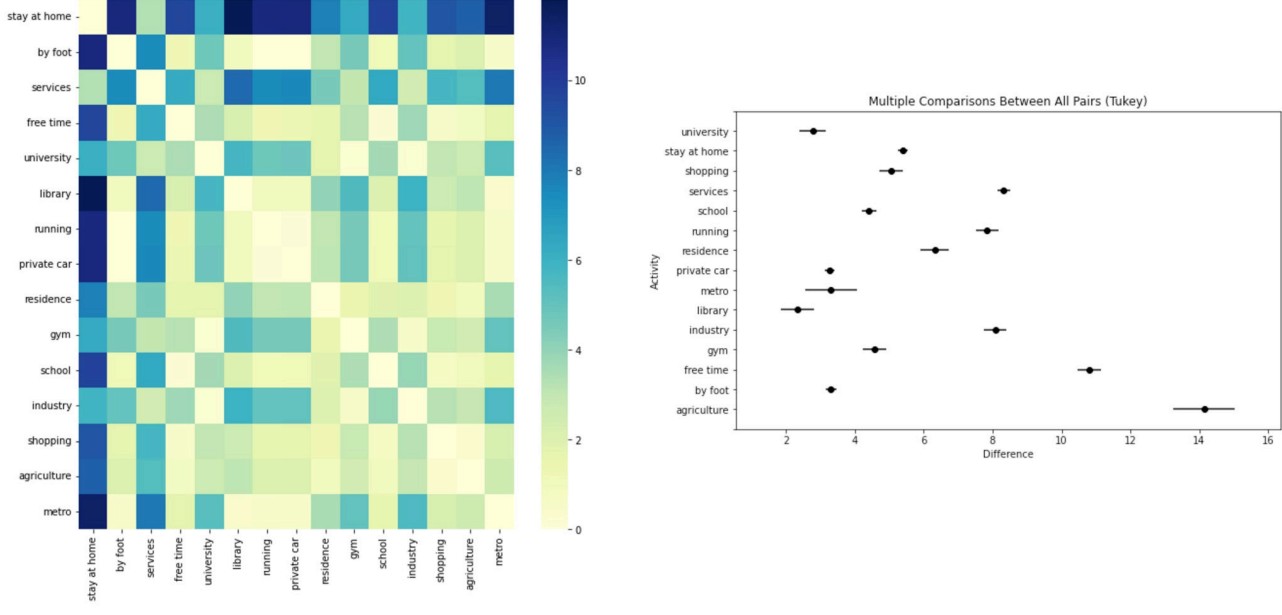

| | SS | DF | F-VALUE | P-VALUE |
|---|---|---|---|---|
| Activity | 3.26e+5 | 14.0 | 739 | 0.0 |
| Residual | 3.19e+6 | 101170 | - | - |

**Fig 5. ANOVA analysis for the mortality relative risk among activities.** The ANOVA table is shown in addition to the heatmap with the mortality relative risk difference between activities and the plot of the Tukey test.

running. A high correlation exists between the activities with high exposure and the activities with an increased relative risk. These results are exposed in Fig 5, which includes the ANOVA table, a heatmap that shows mean differences between activities and the Tukey posthoc analysis.

## Interactive Process Mining

The Interactive Process Mining workflow provides a process map that explains all the daily activities of the population Fig 6. The colour gradient in each node represents the time spent. In this line, the redder the node, the more time spent on the activity. This flow enhancement can illustrate the characteristics of daily activities. In that case, higher times are associated with work activities, around eight hours. Thus, as personal exposure is a function of time, it is possible to interpret that the activities with a high time are related to increased exposure. In contrast, greener activities could be associated with low exposure and risk. In this sense, according to the process map shown in Fig 5, the activities with the highest amount of time spent are staying at home and the work activities—agriculture, services, industry, residence (understood as people that work in another home) and agriculture—and studying activities—university, school and library. On the other hand, the activities that cover low quantities of time are the

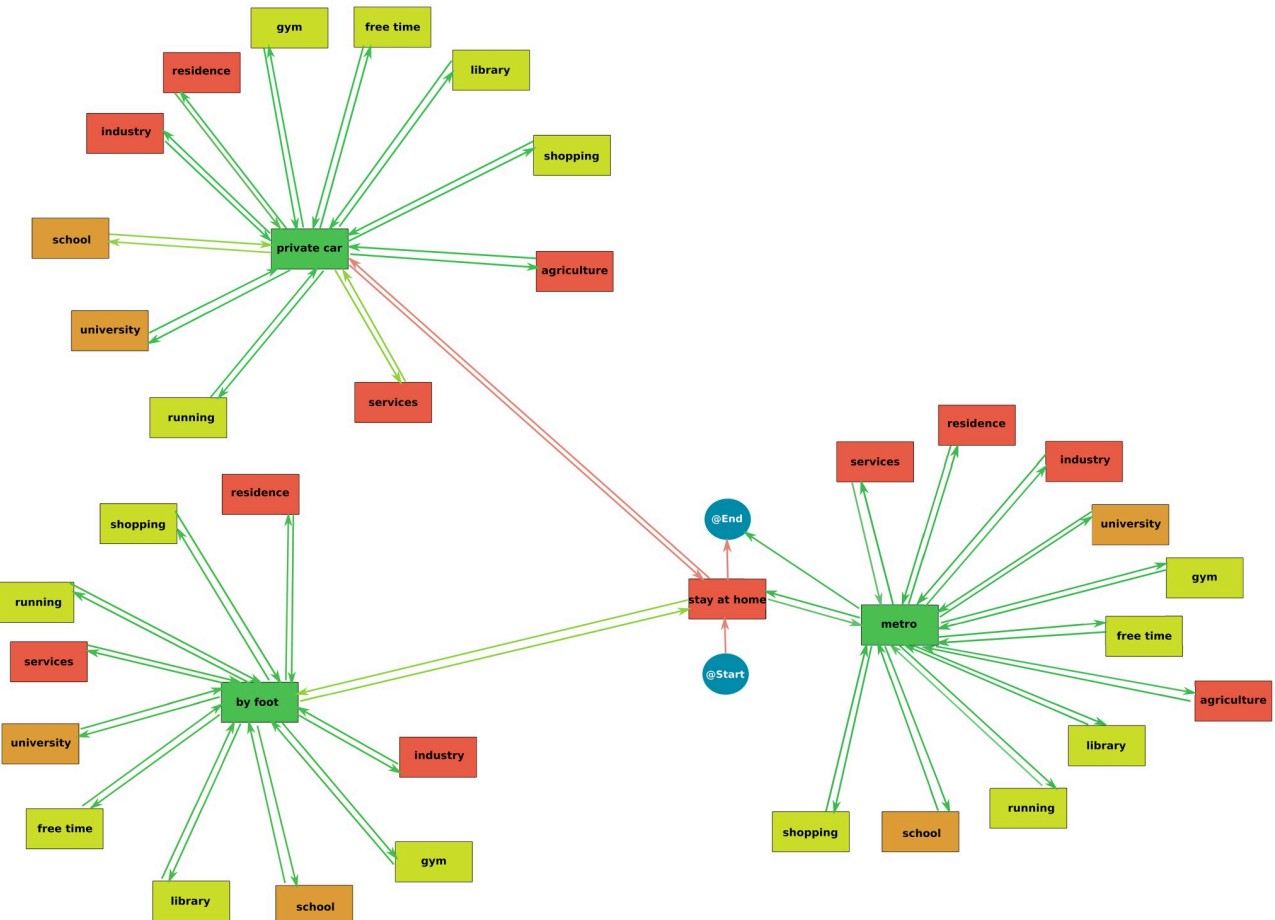

**Fig 6. Process map of the daily activities in the populations.** Redder nodes represent activities with a high spent time and green nodes show activities with a low spent time.

movement activities—by foot, private car and metro—because they depend on the distance, a parameter that in some cases could be small.

However, this process map is not enough to assess target activities. Therefore, it is necessary to compute the process map of the activities of the traces associated with high exposure (Fig 7). When the statistical significance is calculated—nodes highlighted in yellow -it is obtained that the nodes that show a significant difference in the time spent in the high exposure traces are stayed at home, moved by foot and moved by private car. In addition, the running node connected to the private car is also significant. Therefore, this significance assessment reveals a set of target activities. It is important to note that this analysis only shows significant differences, possibly caused by a higher or lower time spent on this activity. To clarify this, it is necessary to compare these Interactive Process Mining results with the results of the ANOVA analysis. According to this methodology, the following three scenarios are proposed: the implementation of an increment in the private car velocity, the improvement of the infrastructures of the residence buildings, the recommendation of safer places to exercise and the regulation of face masks in outdoor and public environments. Then, combinations of the best scenarios will be done to improve the KPIs.

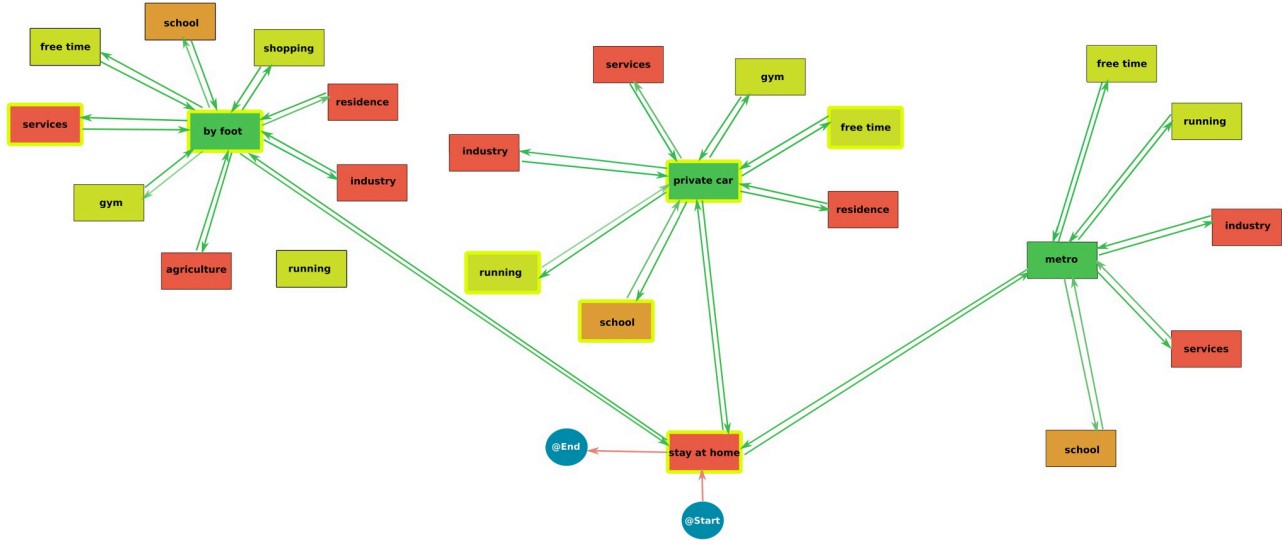

**Fig 7. Process map of the traces (citizens) with high exposures.** Highlighted nodes represent nodes with a significative difference in the spent time in comparison with the process map in Fig 6.

## Scenario 1: Increase private car velocity

The application of the first scenario reaches an improvement in the KPIs, as shown in Fig 8. The 24-hour population exposure is reduced to 118 $\mu g * \text{h}/m^3$, improving this KPI by 1%. In contrast, the population's relative risk remains constant at 1.9. In addition, the percentage of risky activities is reduced to 54% and the number of hours spent on these unsafe activities is the same (3.51 h).

## Scenario 2: Improving building infrastructures

The application of the second scenario achieves a considerable improvement in the KPIs, as shown in Fig 9. The 24-hour population exposure is reduced to 98 $\mu g * \text{h}/m^3$, improving this

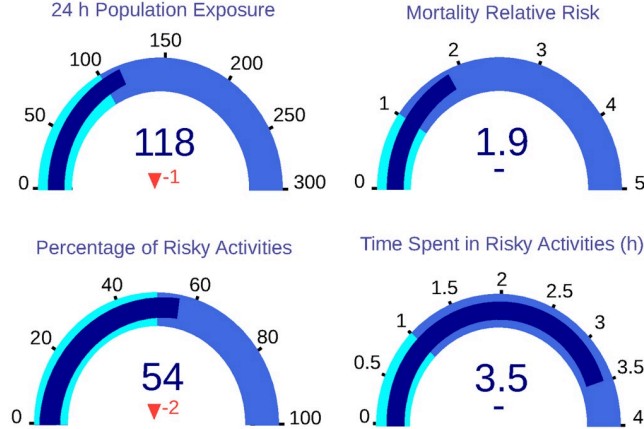

**Fig 8. Dashboard with KPIs for Scenario 1, in which population exposure and percentage of risky activities are reduced.**

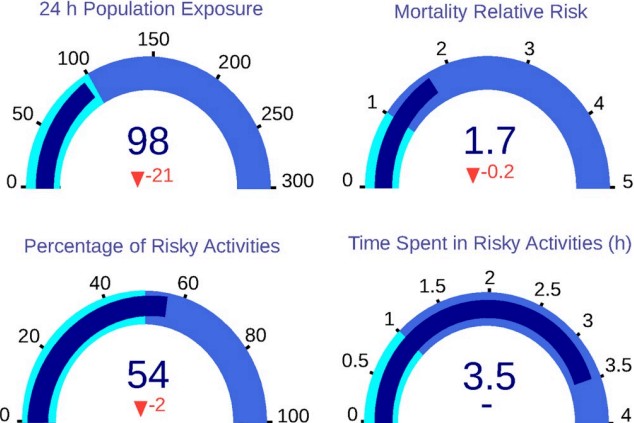

**Fig 9. Dashboard with KPIs for Scenario 2, in which population exposure, mortality relative risk and percentage of risky activities are reduced.**

value an 18%. This effect is also observed in the population relative risk, changing its value from 1.90 to 1.70, which means a 12% improvement. In addition, the percentage of risky activities is reduced to 54%, and the number of hours spent on these dangerous activities remains at 3.5.

## Scenario 3: Recommend safe running places

The application of the third scenario achieves a slight improvement on three of the four KPIs, as it is represented in the gauges (Fig 10). 24-hour population exposure is reduced to 116 $\mu g *$ h/$m^3$, supposing a reduction of 3%. On the other hand, the population relative risk remains at 1.90. In addition, the percentage of risky activities remains at 56%, and the number of hours spent in unsafe activities also remains at 3.50.

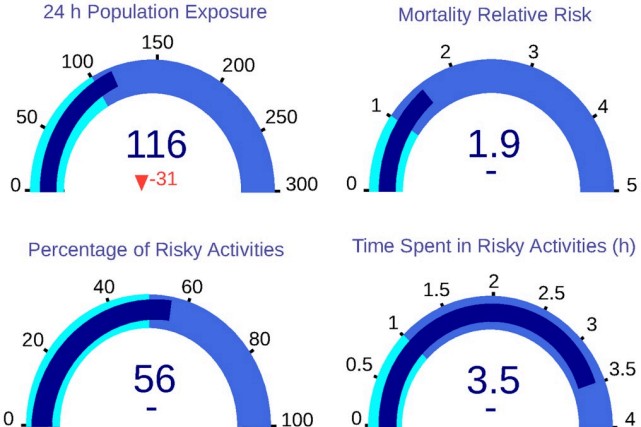

**Fig 10. Dashboard with KPIs for Scenario 3, in which population exposure is reduced.**

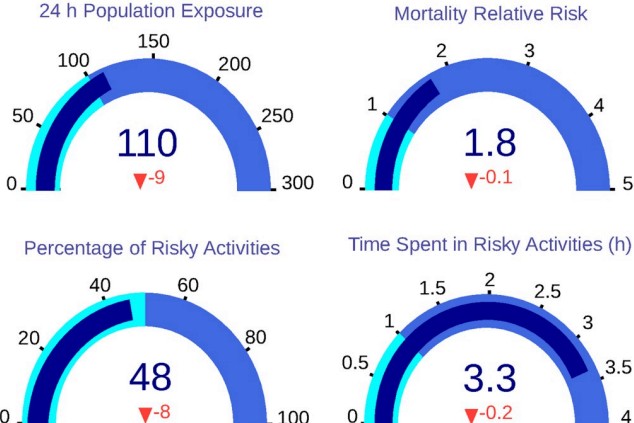

**Fig 11. Dashboard with KPIs for Scenario 4, in which all KPIs are reduced.**

## Scenario 4: Use face masks in outdoor and public environments

This scenario proposes the regulation of using a face mask in outdoor activities and also in public buildings—as universities, schools, libraries and shopping centres. The implementation of these measurements achieves an important improvement in all KPIs, as it is represented in the gauges (Fig 11). 24-hour population exposure is reduced to 110 $\mu g * \text{h}/m^3$, supposing a reduction of 8%. On the other hand, the population's relative risk is reduced to 1.80. Last, the percentage of risky activities decreases to 48%, and the number of hours spent in unsafe activities is reduced to 3.50.

## Scenario 5: Scenario 2 + Scenario 4

The last scenario proposes a combination of the best ones (Scenario 2 + Scenario 4), applying the face mask regulations and the improvement of buildings infrastructures to reduce I/O ratios. Applying this combination of measurements obtains the best results, as shown in Fig 12. 24-hour population exposure is reduced to 88 $\mu g * \text{h}/m^3$, supposing a reduction of 26%. On the

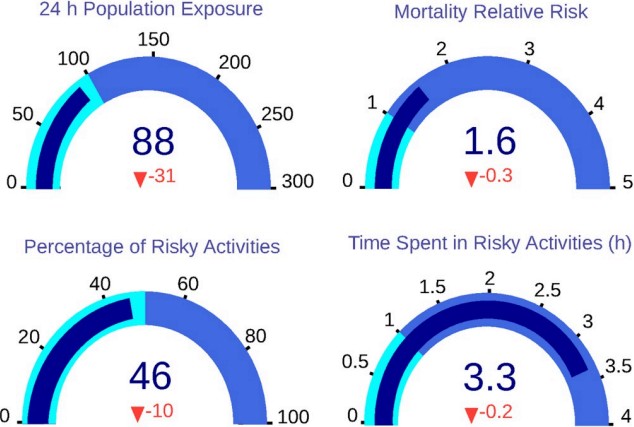

**Fig 12. Dashboard with KPIs for Scenario 5, in which all KPIs are reduced in a greater percentage in comparison with Fig 11.**

**Table 1. KPIs for different iterations of the model.**

|  | Population Exposure | Relative Risk | Risky Activities | Time Spent |
|---|---|---|---|---|
| Scenario 1 | 117 117 116 | 1.9 1.9 1.9 | 54 54 55 | 3.5 3.6 3.5 |
| Scenario 2 | 96 96 96 | 1.7 1.7 1.7 | 55 55 55 | 3.5 3.5 3.5 |
| Scenario 3 | 117 117 116 | 1.9 1.9 1.9 | 55 55 55 | 3.6 3.6 3.5 |
| Scenario 4 | 110 109 110 | 1.8 1.8 1.8 | 48 47 48 | 3.4 3.3 3.3 |
| Scenario 5 | 88 88 88 | 1.6 1.6 1.6 | 46 46 46 | 3.3 3.3 3.3 |

other hand, the population's relative risk decreases a 16% to 1.60. In addition, the percentage of risky activities is reduced to 46%, and the number of hours spent in unsafe activities to 3.30.

## Analysis of model variability

This section analyses the variability of the KPIs when several simulations are performed. This is because the sequences generated in each iteration are different when working with static models. These analyses have been carried out in triplicate for each of the proposed scenarios (Table 1). In general, little variability can be observed, especially in Scenario 2 and Scenario 5, where the same results are obtained for the three experiments. On the other hand, the most sensitive KPI is 24 h population exposure, followed by the Percentage of Risky Activities and Time Spent in Risky Activities. Finally, the Mortality Relative Risk does not change in the different simulations.

## Discussion

The previous results demonstrate that Interactive Process Mining can identify target activities for exposure mitigation, showing that this methodology could perform a sequence-oriented sensitive analysis. This hypothesis gets more robust because the proposed scenarios reduce both population exposure and population risk—in the best case, the improvement of building infrastructures combined with a strong face mask policy reduces exposure by 26% and the mortality relative risk by 16%. The strong combination of these two scenarios is because of their impact in a separate ways. The improvement of building infrastructures helps to reduce population exposure by 18% and relative risk by 12%. In contrast, the use of face masks reduces these two KPIs by only 8% and 5%, but has a great impact on the percentage of risky activities—it is reduced a 14%. Thus, it is possible two conclude that the combinations of both scenarios exploit important synergies that allow for obtaining better KPIs.

These results support that Interactive Process Mining could become a powerful environmental modelling and exposure assessment tool. It is important to note that the scenario related to residence buildings is remarkably successful because of the high time spent at home, as it is shown in the process map, and it is a court node in the graph—all the traces pass by this activity. The other scenarios are less successful because they do not affect all individuals. The time spent and the percentages on risky activities do not change much since the time distribution of the activities—the time people spend in each one- does not change in our scenarios. These conclusions are like the ones obtained in the EXPLUME simulation [8].

This sequence-oriented sensitive analysis presents several advantages concerning the classical methodologies. The main strength is sequence analysis and sequence data, because experts in interactive process mining can do a track. The analysis without edges between nodes can be extracted for ANOVA results, and it is observed that they are similar, but the strength here is that the results are more explainable. For instance, by studying the sequence track it is possible

to conclude that acting on building infrastructure is quite effective because it is a pattern that appears in all the sequences, and the same applies to studying less successful scenarios. Thus, the interactive process mining workflow is more agile, explainable, and human-oriented, avoiding the black box effect. Clearly, this new methodology shows the target parameters, and it is only necessary to iterate on these few inputs. On the other hand, the traditional approach iterates over all the parameters to find the best combination. However, this hypothesis should be tested by conducting a pilot with real data. This approach allows extrapolation of our method for forecasting purposes, using chemistry transport models to generate predictions in gridded domains [16], being possible to perform this analysis for other pollutants, as it is done in Paris [8]. Furthermore, the most critical point is that sequence-oriented sensitive analysis is a more intelligent approach because it is based on understandable models and human criteria. However, this methodology has some potential limitations: it is possible to miss some crucial parameters in the exposure contribution, and the outcomes are susceptible to wrong human interpretations.

In this paper, the results obtained for the population generated cannot be extrapolated to the actual population of Valencia. Logically, since the model is based on accurate statistics, the synthetic dataset must be representative and have some correlation. In this work, we can ensure that the data is representative of the overall population. We have used the most recent statistics, and since the number of individuals in our sample is high (100856 activities and 15795 individuals), we can assume that the distribution of the parameters is similar to the real one, according to the central limit theorem. On the other hand, dynamic data is generated using the activities and movement statistics for the city of Valencia—previous work has validated this methodology [8]. For this reason, the results presented in this paper should be interpreted as a proof of concept of the proposed methodology and mode of use. In other words, our study aims to demonstrate the validity of our method for further steps and studies.

However, this synthetic data is reliable to confirm our hypothesis. The first reason is observed in Table 1, where all the simulations compute the same values for the four KPIs in each scenario. There is a little variation in some decimals but no significant differences. This confirms that our method is robust despite it being stochastic, and the sample generated is representative of the same population. Of course, this population could be different from the real one, highlighting the necessity of a real pilot.

In this sense, the rise of wearable smartwatches that allows fitness and sleep activity monitoring has created a cultural phenomenon called the quantified self, whereby members of the general population voluntarily wear tracking devices that continuously log their data in exchange for potential improvements in the quality of life or physical performance. This paradigm can be applied to the real-time monitoring of air pollution. These sensors have been tested in the canarin project for medical purposes [39], enabling people to change their behaviour or avoid pollution. Several studies have demonstrated that citizen engagement against air pollution is low, close to 12% [40]. Thus, the next step is to apply Interactive Process Mining to real data and obtain conclusions for the real population. The emergence of the IoT paradigm allows the creation of architectures in which it is easy to integrate all these processes (Fig 13). In addition, some pilots that combine mobile monitoring and modelling have demonstrated that they can reduce costs and improve accuracy [14].

Last, Interactive Process Mining requires multidisciplinary since the generated models and process maps must be validated by environmental experts. This fact has already been observed in Interactive Process Mining applied to healthcare, where the role of data rodeos in coordination with clinicians is essential [37]. In this sense, a similar methodology should be applied to the sequence-oriented sensitive analysis. As a result, the sequence-oriented sensitive analysis is a paradigm that involves urban planning professionals in the middle of understanding the

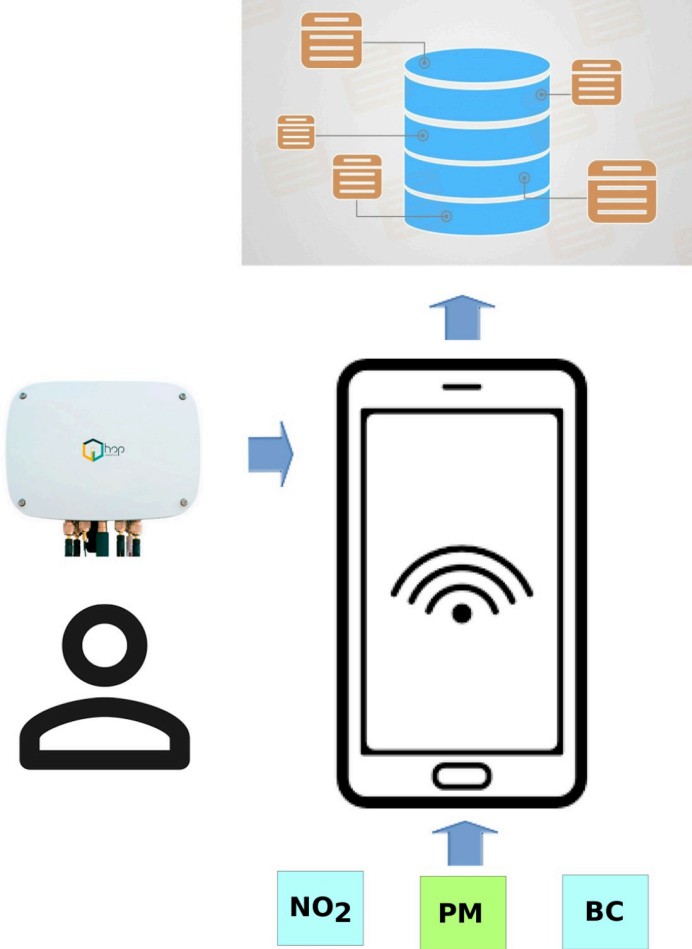

**Fig 13. IoT architecture.**

process until a reliable KPI is defined, which can be computed from the data available in the system and through iterative data rodeos sessions. In addition, the expert should participate in the process map interpretation and scenario assessment. This last step is critical because the effect on the KPIs could be missed without defining a good scenario.

## Conclusion

In conclusion, this work has developed a methodology based on Interactive Process Mining to perform a sequence-oriented sensitive analysis, demonstrating that the use of Interactive Process Mining substantially improves the identification of target activities related to high exposure and allows experts to take the best urban policies. Moreover, the results were obtained concerning the KPIs, with a reduction of 26% in the PM2.5 exposure and a decrease of 16% in the mortality relative risk in the best scenario. Thus, the results suggest that our initial hypothesis is correct, and the sequence-oriented sensitive analysis could be a powerful tool to improve the environment in cities.

On the other hand, the results obtained open up new research possibilities, which can be classified along with three different aspects: i) testing the methodology with real exposure data,

seeking to understand which activities are related to real exposure, and using the architectures and infrastructures of the emerging IoT paradigm; ii) integrating urban experts in the Interactive Process Mining workflow, in the definition of KPIs as well as in the suggestion of scenarios and iii) improving the workflow adding new test and analysis that can complement the results. On the other hand, the use of the methodology proposed allows progress in the following lines that have been limited so far in state of the art: i) the assessment of public health policies to reduce population exposure; ii) the improvement of the knowledge in the relation of exposure and risk, enabling researchers to define accurate exposure-response models and iii) the applications of Interactive Process Mining in environmental modelling and air pollution mitigation.

## Supporting information

**S1 Data. Air quality data.** Gridded Air Quality Data for the city of Valencia. The grid is divided in several cells, in which one is computed the PM2.5 concentration. It is composed by the following variables: *cell_id* (the identifier for each cell in the grid), *PM2.5* (PM2.5 concentration in $\mu g/m_3$), *datetime* (date in utc format), *hourId* and *dayId*. **File: ./data/aq_data/valencia_aq_gridded_v2.csv.**
(CSV)

**S2 Data. Population data.** Data with each one of the individuals in the populations, as well as the epidemiological and social data. It is composed by the following variables: *id* (citizen identifier), *sex*, *age*, *obesity*, *diabetes*, *asthma*, *high blood pressure*, *pulmonar disease*, *heart disease*, *anxiety*, *smoke*, *alcohol* (binary epidemiological variables, if the value is 1 the citizen suffer the affection), *employ*, *transport*, *freeTime*, *residence* (indicate the identifier of the employ, transport, freeTime and residence locations in the geoJSON file). **File: ./data/population_data/sample_population_v3.csv.**
(CSV)

**S3 Data. Base scenario sequences.** Data with the activities performed by citizens in a day, with the corresponding locations, exposure and risk for the base population. It is composed by the following variables: *id* (citizen identifier), *cell_id* (the identifier for each cell in the grid, activities related to two or more cells are coding as the concatenation of the cell_id of all the included cells), *order*, *Activity*, *Duration*, *DateStart*, *DateEnd*, *Sex*, *Age*, *Asthma*, *Diabetes*, *High Blood Pressure*, *Pulmonar Disease*, *Heart Disease*, *Anxiety*, *Smoke*, *Alcohol* (binary epidemiological variables, if the value is 1 the citizen suffer the affection), *concentration* (PM2.5 concentration in $\mu g * min/m_3$), *level*, *mortality relative risk*. **File: ./data/sequence_data/sintetic_data_v6.csv.**
(CSV)

**S4 Data. Scenario 1 sequences.** Data with the activities performed by citizens in a day, with the corresponding locations, exposure and risk for the first scenario population. It is composed of the following variables: *id* (citizen identifier), *cell_id* (the identifier for each cell in the grid, activities related to two or more cells are coding as the concatenation of the cell_id of all the included cells), *order*, *Activity*, *Duration*, *DateStart*, *DateEnd*, *Sex*, *Age*, *Asthma*, *Diabetes*, *High Blood Pressure*, *Pulmonar Disease*, *Heart Disease*, *Anxiety*, *Smoke*, *Alcohol* (binary epidemiological variables, if the value is 1 the citizen suffer the affection), *concentration* (PM2.5 concentration in $\mu g/m_3$), *exposure* (PM2.5 expsure in $\mu g * min/m_3$), *level*, *mortality relative risk*. **File: ./data/sequence_data/sintetic_data_v6_car.csv.**
(CSV)

**S5 Data. Scenario 2 sequences.** Data with the activities performed by citizens in a day, with the corresponding locations, exposure and risk for the second scenario population. It is composed of the following variables: *id* (citizen identifier), *cell_id* (the identifier for each cell in the grid, activities related to two or more cells are coding as the concatenation of the cell_id of all the included cells), *order*, *Activity*, *Duration*, *DateStart*, *DateEnd*, *Sex*, *Age*, *Asthma*, *Diabetes*, *High Blood Pressure*, *Pulmonar Disease*, *Heart Disease*, *Anxiety*, *Smoke*, *Alcohol* (binary epidemiological variables, if the value is 1 the citizen suffer the affection), *concentration* (PM2.5 concentration in $\mu g/m_3$), *exposure* (PM2.5 expsure in $\mu g * min/m_3$), *level*, *mortality relative risk*.
**File: ./data/sequence_data/sintetic_data_v6_residence.csv.**
(CSV)

**S6 Data. Scenario 3 sequences.** Data with the activities performed by citizens in a day, with the corresponding locations, exposure and risk for the third scenario population. It is composed by the following variables: *id* (citizen identifier), *cell_id* (the identifier for each cell in the grid, activities related to two or more cells are coding as the concatenation of the cell_id of all the included cells), *order*, *Activity*, *Duration*, *DateStart*, *DateEnd*, *Sex*, *Age*, *Asthma*, *Diabetes*, *High Blood Pressure*, *Pulmonar Disease*, *Heart Disease*, *Anxiety*, *Smoke*, *Alcohol* (binary epidemiological variables, if the value is 1 the citizen suffer the affection), *concentration* (PM2.5 concentration in $\mu g/m_3$), *exposure* (PM2.5 expsure in $\mu g * min/m_3$), *level*, *mortality relative risk*.
**File: ./data/sequence_data/sintetic_data_v6_running.csv.**
(CSV)

**S7 Data. Scenario 4 sequences.** Data with the activities performed by citizens in a day, with the corresponding locations, exposure and risk for the fourth scenario population. It is composed by the following variables: *id* (citizen identifier), *cell_id* (the identifier for each cell in the grid, activities related to two or more cells are coding as the concatenation of the cell_id of all the included cells), *order*, *Activity*, *Duration*, *DateStart*, *DateEnd*, *Sex*, *Age*, *Asthma*, *Diabetes*, *High Blood Pressure*, *Pulmonar Disease*, *Heart Disease*, *Anxiety*, *Smoke*, *Alcohol* (binary epidemiological variables, if the value is 1 the citizen suffer the affection), *concentration* (PM2.5 concentration in $\mu g/m_3$), *exposure* (PM2.5 expsure in $\mu g * min/m_3$), *level*, *mortality relative risk*.
**File: ./data/sequence_data/sintetic_data_v6_mask.csv.**
(CSV)

**S8 Data. Scenario 5 sequences.** Data with the activities performed by citizens in a day, with the corresponding locations, exposure and risk for the fifth scenario population. It is composed by the following variables: *id* (citizen identifier), *cell_id* (the identifier for each cell in the grid, activities related to two or more cells are coding as the concatenation of the cell_id of all the included cells), *order*, *Activity*, *Duration*, *DateStart*, *DateEnd*, *Sex*, *Age*, *Asthma*, *Diabetes*, *High Blood Pressure*, *Pulmonar Disease*, *Heart Disease*, *Anxiety*, *Smoke*, *Alcohol* (binary epidemiological variables, if the value is 1 the citizen suffer the affection), *concentration* (PM2.5 concentration in $\mu g/m_3$), *exposure* (PM2.5 expsure in $\mu g * min/m_3$), *level*, *mortality relative risk*.
**File: ./data/sequence_data/sintetic_data_v6_scenario5.csv.**
(CSV)

## Acknowledgments

### Code availability

Source codes are available on the GitHub repository (https://doi.org/10.5281/zenodo.8079155). This includes the scripts to generate the synthetic data and compute the KPIs. The

PMApp tool has been developed in the context of the VALUE project. It should be requested from the authors (https://valueproject.eu/)

## Author Contributions

**Conceptualization:** Eduardo Illueca Fernández, Carlos Fernández Llatas, Fernando Seoane Martinez.

**Data curation:** Eduardo Illueca Fernández, Fernando Seoane Martinez.

**Formal analysis:** Eduardo Illueca Fernández, Carlos Fernández Llatas, Jesualdo Tomás Fernández Breis, Fernando Seoane Martinez.

**Funding acquisition:** Fernando Seoane Martinez.

**Investigation:** Eduardo Illueca Fernández, Carlos Fernández Llatas.

**Methodology:** Eduardo Illueca Fernández, Fernando Seoane Martinez.

**Software:** Eduardo Illueca Fernández.

**Supervision:** Antonio Jesús Jara Valera, Jesualdo Tomás Fernández Breis, Fernando Seoane Martinez.

**Validation:** Jesualdo Tomás Fernández Breis, Fernando Seoane Martinez.

**Writing – original draft:** Eduardo Illueca Fernández, Carlos Fernández Llatas.

**Writing – review & editing:** Eduardo Illueca Fernández, Carlos Fernández Llatas, Antonio Jesús Jara Valera, Jesualdo Tomás Fernández Breis, Fernando Seoane Martinez.

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
