## [Decision Letter · Decision Letter 0]

30 May 2023

PONE-D-23-03914Sequence-Oriented Sensitive Analysis for PM2.5 exposure and risk assessment using Interactive Process MiningPLOS ONE

Dear Dr. Illueca Fernandez,

Thank you for submitting your manuscript to PLOS ONE. After careful consideration, we feel that it has merit but does not fully meet PLOS ONE’s publication criteria as it currently stands. Therefore, we invite you to submit a revised version of the manuscript that addresses the points raised during the review process.

We look forward to receiving your revised manuscript.

Kind regards,

Sathishkumar V E

Academic Editor

PLOS ONE

Journal Requirements:

This work has been supported by the fellowship 21300/FPI/19 funded by Fundaci´on 556

S´eneca and co-funded by HOP Ubiquitous S.L. Regi´on de Murcia (Spain), grant Nº 557

21681/EFPI/21. This activity has received funding from EIT Health (www.eithealth.eu) 558

ID 220649, the innovation community on Health of the European Institute of Innovation 559

and Technology (EIT), a body of the European Union, under Horizon 2020, the EU 560

Framework Programme for Research and Innovation. 

The author EIF has received funded from Fundacion Séneca (https://fseneca.es/), grant number 21300/FPI/19

The authors have received funded from EIT Health (https://eithealth.eu/), grant number 220649

The funders had no role in study design, data collection and analysis, decision to publish, or preparation of the manuscript

Reviewers' comments:

Reviewer's Responses to Questions

**Comments to the Author**

1. Is the manuscript technically sound, and do the data support the conclusions?

Reviewer #1: Yes

Reviewer #2: Yes

2. Has the statistical analysis been performed appropriately and rigorously? 

Reviewer #1: I Don't Know

Reviewer #2: Yes

3. Have the authors made all data underlying the findings in their manuscript fully available?

Reviewer #1: Yes

Reviewer #2: Yes

4. Is the manuscript presented in an intelligible fashion and written in standard English?

Reviewer #1: Yes

Reviewer #2: Yes

5. Review Comments to the Author

Reviewer #1: The authors present an approach to identify exposure-related activities in the context of PM2.5. The approach is sequence-oriented and based, among other things, on the use of Interactive Process Mining. The paper begins with an introduction to PM2.5, the health effects and current approaches and studies PM2.5, the concentration of PM2.5 at specific locations, and the limitation of previous studies because they do not include the factor of time. The authors then explain the state of the art in the area of health impact of particle matter and its relationship with the exposure population. After that, the authors describe their procedure, which includes the generation of the synthetic data on which the evaluation will be based later, the actual sequence-oriented sensitive analysis and the calculation of the KPIs. At the end, the data set, the ANOVA analysis and the Interactive Process Mining and the results are described.

The paper is very well written and the authors have a lot of knowledge in the topic PM2.5 and the state of the art. However, the paper has some weaknesses, the correction of which would improve the paper significantly.

(1) The role of sequence in the analysis is not made clear. What are the implications of sequence and sequential consideration of activities for the overall goal? What are the advantages of sequential representation? Would color highlighting in this way, without edges between nodes lead to similar results?

(2) If I understood correctly, interactive process mining was performed. This always involves experts who evaluate the process models? It is not clear how exactly the evaluation was done. Who carried out the interactive process mining? How was it evaluated? Who derived the suggestions? How was the experiment conducted?

(3) Basics: the understanding of the paper could be significantly supported by explaining basics (either in a separate basics chapter or in an existing chapter).

Minor issues and open questions:

- Figures are partially pixelated and not readable

- The figures should be better described. Percentages at edges in process models were not clear.

- Why do activities appear multiple times in the process models (industry, for example)?

- What were the reasons that led to the selection of the PALIA algorithm?

- A figure describing the process of application and evaluation would facilitate understanding.

- Typo in "oriented sensitive analysis" -> sequention-oriented? semiautomatic -> semi-automatic? adition -> addition? Partially inconsistent British/American English. expsure

It is very interesting to bring techniques from process mining into other domains and possibly (so far) not typical application areas.

Reviewer #2: Good work done for the benefit of the environment using a new tool with potential benefits for different local systems. Very interesting document with important data, adequate statistical validation but presented in a very understandable way. Congratulations.

6. PLOS authors have the option to publish the peer review history of their article (what does this mean?). If published, this will include your full peer review and any attached files.

Reviewer #1: No

Reviewer #2: No

---

## [Author Response · Author response to Decision Letter 0]

2 Jul 2023

We thank the reviewers for their valuable comments, which have contributed to improving the quality of the manuscript. The suggestions and comments have been closely followed, and revisions have been made accordingly, which we hope meets your expectations. 

Please, find below the comments of the Reviewers and our responses inserted after each comment. The additions to the manuscript text have been highlighted in italic letters because more text is provided to keep the context. Please note that figure, paragraph, and line numbers refer to the manuscript with track changes, not the original one. The response to the editorial comments is also included at the end of the response.

Reviewer #1

Q1. The role of sequence in the analysis is not made clear. What are the implications of sequence and sequential consideration of activities for the overall goal? What are the advantages of sequential representation? Would color highlighting in this way, without edges between nodes lead to similar results? 

R1. We agree with the reviewer that the role of sequence in the analysis could be explained in more detail. The key point is that sequence analysis considers each activity's contribution and allows one to compute the exposure per citizen. This allows for avoiding population metrics that could be biased by the sampling process - this can also happen in sequential analysis. Still, experts in interactive process mining can do a track. The analysis “without edges between nodes” can be extracted for ANOVA results, and it is observed that they are similar, but the strength here is that the results are more explainable. Thus, the interactive process mining workflow is more agile.

To clarify this, the following sentences are added to the introduction.

“In our view, this limits the model considerably and leads to a loss of information because it is not possible to track how pollution affects each individual, and it is not possible to compute individual KPIs. Thus, sequential data and sequence-oriented analysis are critical to overcome these limitations. ”

The following explanation has been added to the Sequence-Oriented Sensitive Analysis subsection, second paragraph.

“The main goal of this work is to perform a sensitivity analysis to reduce the exposure and the health outcomes by taking action on the activities. In other words, the main goal is to identify which activities contribute more to population exposure. Sequence analysis is essential for our proposed optimisation because this allows for avoiding population metrics that the sampling process could bias. Thus, it allows for performing analyses for different population sectors - for instance, analysing only the persons affected by cardiac issues - or even tracking and optimising the activities of a single individual [1]. In addition, the classical sensitive analysis should be performed by doing different simulations among the inputs to find the inputs that minimise a set of KPIs. The computational cost of this sensitive analysis could be high due to the number of input parameters to consider. Thus, we propose a new methodology to assess an oriented sensitive analysis using interactive process mining (IPM) in this paper.”

Last, the following contribution was added to the discussion, in concrete, in the third paragraph.

“This sequence-oriented sensitive analysis presents several advantages concerning the classical methodologies. The main strength is sequence analysis and sequence data because experts in interactive process mining can do a track. The analysis without edges between nodes can be extracted for ANOVA results, and it is observed that they are similar, but the strength here is that the results are more explainable. For instance, by studying the sequence track it is possible to conclude that acting on building infrastructure is quite effective because it is a pattern that appears in all the sequences. The same applies to studying less successful scenarios. Thus, the interactive process mining workflow is more agile, explainable, and human-oriented, avoiding the black box effect. Clearly, this new methodology shows the target parameters, and it is only necessary to iterate on these few inputs. On the other hand, the traditional approach iterates over all the parameters to find the best combination. However, this hypothesis should be tested by conducting a pilot with real data. This approach allows extrapolation of our method for forecasting purposes, using chemistry transport models to generate predictions in gridded domains [2], being possible to perform this analysis for other pollutants, as it is done in Paris [3]. Furthermore, the most critical point is that sequence-oriented sensitive analysis is a more intelligent approach based on understandable models and human criteria. However, this methodology has some potential limitations: it is possible to miss some crucial parameters in the exposure contribution, and the outcomes are susceptible to wrong human interpretations.”

[1] Zhang L, Guo C, Jia X, Xu H, Pan M, et al. (2018) Personal exposure measurements of school-children to fine particulate matter (PM2.5) in winter of 2013, Shanghai, China. PLOS ONE 13(4): e0193586. https://doi.org/10.1371/journal.pone.0193586

[2] Menut L, Bessagnet B, Khvorostyanov D, Beekmann M, Blond N, Colette A, et al. CHIMERE 2013: a model for regional atmospheric composition modelling. Geoscientific model development. 2013;6(4):981–1028.

[3] Valari M, Markakis K, Powaga E, Collignan B, Perrussel O. EXPLUME v1. 0: a model for personal exposure to ambient O3 and PM2.5. Geoscientific Model Development. 2020;13(3):1075–1094.

Q2. If I understood correctly, interactive process mining was performed. This always involves experts who evaluate the process models? It is not clear how exactly the evaluation was done. Who carried out the interactive process mining? How was it evaluated? Who derived the suggestions? How was the experiment conducted?

R2. We agree with the reviewer that interactive process mining always involves an evaluation from a team. This has been done indeed in this work. The approach has been validated with urban planners experts from Libelium, an IoT company that works on sustainability impact assessment in Smart Cities. The interactive process mining has been performed by Eduardo Illueca Fernández (UM) and Carlos Fernández Llatas (ITACA-SABIEN), supervised by Fernando Seoane Martinez (KI). Antonio Jesús Jara Valera (Libelium) derived the suggestions for the target scenario as an expert in sustainability impact assessment, and the validation has been performed by computing the KPIs again and obtaining improvements.

To clarify this, the following paragraph (third paragraph) and Figure 2 have been added to the manuscript.

“Interactive process mining is a methodology that requires an iterative human validation by domain experts through the so-called data rodeo. In this way, the domain experts can i) iteratively define a process indicator according to the sensitive analysis goals, ii) analyse and validate the process indicator and iii) be trained in using the process indicator. This process is oriented by an interactive process indicator (IPI) - in this case, IPI can be personal exposure or mortality relative risk. Once an IPI is defined, the domain expert is ready to analyse the data using an interactive process mining tool. This process is repeated until reaching the target scenarios, which should be validated by computing the KPIs again and checking for achieving the optimisation goal, in this case, reducing personal exposure and relative risk. In this work, the involved stakeholders are Libelium as domain experts - an IoT company that provides sustainability impact assessment solutions in smart cities - and data scientists from Karolinska Institutet, ITACA-Sabien, and the University of Murcia. The interactive process mining workflow is summarised in Figure 2. [4] ”

Figure 2. The first step of the IP is the preparation phase, where the simulations are performed; then, the generated data is processed through interactive process mining in constant discussion with domains expert, and the proposed scenarios are validated with domain experts once the KPIs are calculated again.

[4] Lull JJ, Bayo JL, Shirali M, Ghassemian M, Fernandez-Llatas C. Interactive process mining in IOT and human behaviour modelling. Interactive process mining in healthcare. 2021; p. 217–231

Q3. Basics: the understanding of the paper could be significantly supported by explaining basics (either in a separate basics chapter or in an existing chapter).

R3. We agree with the reviewer that explaining basic concepts can help readers understand the work. For this reason, we have added a first subsection in the Materials and Methods chapter, Fundamentals and Basic Concepts, that we attach hereafter. 

“The atmosphere, whether urban or remote areas, contains many aerosol particles suspended. From a physical point of view, particle matter (PM) is a mixture of solid particles and liquid droplets found in the air. Some particles, such as dust, dirt, soot, or smoke, are large or dark enough to be seen with the naked eye. Others are so small they can only be detected using an electron microscope. This wide size range can be appreciated by considering that the mass for one 10 μm is equivalent to the mass of one billion 10 nm particles. Thus, working with each particle as a single entity is difficult. It is necessary to work with particle populations characterised by a cumulative size distribution, defined as the particles that are smaller than or equal to this size range. From this cumulative distribution, the concepts PM10, PM2.5, and PM1 arise, which are particles smaller than or equal to 10 μm, 2.5 μm, and 1 μm, respectively [5].

This work focuses on the PM2.5 population - also called the PM2.5 fraction - because of the impact on health. A percentage of a particle population can pass through the alveolar barrier in the lungs and reach the blood torrent. This percentage of particles is the respirable fraction responsible for adverse health effects and premature deaths. The size of these respirable particles depends on their chemical composition and the corporal weight of the individual. The percentage of particles smaller than PM2.5 that can pass the alveolar barrier is close to 100 %, showing the need to focus mitigation policies on this dangerous pollutant [6]. 

For this elaborate effective strategy, it is necessary to measure the impact of a concentration of PM2.5 in the air. This is done by computing KPIs. The exposure is the closest to concentration, and it has been previously defined as the number of particles a citizen is exposed to in an interval of time. This amount of pollutant affects the individual and can be modelled as the effect of other drugs or toxic. In this sense, concentration-response functions act as a model that measures this effect in a concrete outcome - generally, this is a pathology. There are a lot of concentration-response functions that link PM2.5 with several health issues, but this work will focus on the effect of PM2.5 on premature mortality [7]. ”

[5] Seinfeld JH, Pandis SN. Properties of atmospheric aerosols. Atmospheric Chemistry and Physics: From Air Pollution to Climate Change. 2006.

[6] Görner P, Simon X, Bémer D, Lidés G. Workplace aerosol mass concentration measurement using optical particle counters. Journal of Environmental Monitoring. 2011; pp 310–317.

[7] Heroux ME, Anderson HR, Atkinson R, Brunekreef B, Cohen A, Forastiere F, et al. Quantifying the health impacts of ambient air pollutants: recommendations of a WHO/Europe project. International journal of public health. 2015;60:619–627.

Q4. Figures are partially pixelated and not readable

R4. We would like to thank the reviewer for highlighting this, and all the figures in the manuscript have been re-edited and modified to guarantee high-quality resolution by checking with Preflight Analysis and Conversion Engine (PACE) digital diagnostic tool, https://pacev2.apexcovantage.com/, suggested by PLOS ONE. We also noted that uploading figures to supplementary materials, the files for Figure 5 and Figure 6 were duplicated (Figure 6 was uploaded twice), and we have corrected this now. 

Q5. The figures should be better described. Percentages at edges in process models were not clear.

R5. We agree with the reviewer that a more exhaustive description could be added to the figures. The manuscript has been updated with these extended captions. Percentages at edges in process maps were removed to improve the quality of the Figure.

Q6. What were the reasons that led to the selection of the PALIA algorithm?

R6. We agree with the reviewer that the reasons for selecting PALIA are not justified in the manuscript. In general terms, we chose the PALIA algorithm after a bibliographic analysis, and we found that this algorithm fits better with the work’s goals. Some sentences have been added in the fourth paragraph in Sequence-Oriented Sensitive Analysis subsection to clarify this:

“In this work, we use the PALIA algorithm, implemented by the Institute of Information and Communication Technologies (ITACA) of the Universidad Politecnica de Valencia, Valencia, Spain. This algorithm is the most appropriate one for our goals, because it is based on activity-based possess mining and produces explainable process maps [8]. In addition, it performs better, in terms of efficacy, than other process mining algorithms, such as heuristic miner [9] or genetic process mining [10]. ”

[8] Fernandez-Llatas C, Valdivieso B, Traver V, Benedi JM. Using process mining for automatic support of clinical pathways design. Data mining in clinical medicine. 2015; p. 79–88.

[9] Weijters AJMM, Ribeiro JTS. Flexible heuristics miner (FHM). In: 2011 IEEE symposium on computational intelligence and data mining (CIDM). 2011; pp 310–317.

[10] de Medeiros AKA, Weijters AJMM, van der Aalst WMP. Genetic process mining: an experimental evaluation. Data Min Knowl Discov. 2007; 14(2):245–304

Q7. A figure describing the process of application and evaluation would facilitate understanding.

R7. We agree with the reviewer that an additional figure will clarify the process of application and evaluation of interactive process mining. For this reason, we have added a new Figure 2 in Sequence-Oriented Sensitive Analysis subsection, as it is explained in more detail in Q1.2.

Q8. Typo in "oriented sensitive analysis" -> sequention-oriented? semiautomatic -> semi-automatic? adition -> addition? Partially inconsistent British/American English. expsure

R8. We thank the reviewer for highlighting these typos, which have been corrected to ensure consistent language in the manuscript. 

Reviewer #2

This reviewer did not ask for changes in the manuscript. 

Editorial comments

QE1. Please ensure that your manuscript meets PLOS ONE's style requirements, including those for file naming. The PLOS ONE style templates can be found at 

RE1. We would like to thank the editor for highlighting the importance of format requirements. We have reviewed the format of our manuscript carefully and applied changes when necessary.

QE2. Thank you for stating the following in the Acknowledgments Section of your manuscript: 

We note that you have provided funding information that is not currently declared in your Funding Statement. However, funding information should not appear in the Acknowledgments section or other areas of your manuscript. We will only publish funding information present in the Funding Statement section of the online submission form. Please remove any funding-related text from the manuscript and let us know how you would like to update your Funding Statement. Currently, your Funding Statement reads as follows: 

The author EIF has received funded from Fundacion Séneca (https://fseneca.es/), grant number 21300/FPI/19. The authors have received funded from EIT Health (https://eithealth.eu/), grant number 220649. The funders had no role in study design, data collection and analysis, decision to publish, or preparation of the manuscript

RE2. We would like to thank the editor for noting this, and we have removed the acknowledgement section and moved the funding information to the cover letter, according to your instructions. The following lines have been added to the cover letter.

The author EIF has received funding from Fundacion Séneca (https://fseneca.es/), grant number 21300/FPI/19. The authors have received funding from EIT Health (https://eithealth.eu/), grant number 220649. The funders had no role in study design, data collection and analysis, the decision to publish, or the preparation of the manuscript

QE3. We note that you have stated that you will provide repository information for your data at acceptance. Should your manuscript be accepted for publication, we will hold it until you provide the relevant accession numbers or DOIs necessary to access your data. If you wish to make changes to your Data Availability statement, please describe these changes in your cover letter and we will update your Data Availability statement to reflect the information you provide.

RE3. Dear Editor, we have uploaded our data and code into a repository in Zenodo that is linked with the following digital identifier object: https://doi.org/10.5281/zenodo.8079155 [11], and it is indexed in OpenAIR. This information has been updated in the manuscript in the code availability section and the cover letter, as you suggested in your feedback.

[11] Illueca Fernández E, Fernandez Llatas C, Jara Valera AJ, Fernández Breis JT, Seoane Martinez F. Sequence Oriented Process Mining (v1.0.0). Zenodo. 2023; https://doi.org/10.5281/zenodo.8079155

QE4. Please review your reference list to ensure that it is complete and correct. If you have cited papers that have been retracted, please include the rationale for doing so in the manuscript text, or remove these references and replace them with relevant current references. Any changes to the reference list should be mentioned in the rebuttal letter that accompanies your revised manuscript. If you need to cite a retracted article, indicate the article’s retracted status in the References list and also include a citation and full reference for the retraction notice.

RE4. Dear editor, we have performed an exhaustive revision of the whole bibliography to check that there are no retracted articles. To clarify this, DOIs have been added to all the references when possible. The new articles added to the manuscript have been specified in this rebuttal letter in their corresponding answer. The following paper has been removed from the manuscript as it does not have a consistent DOI

Olsson D, Brabäck L, Forsberg B. Air pollution exposure during pregnancy and infancy and childhood asthma. European Respiratory Journal. 2014;44(Suppl 58).

In addition, the reference “of Disease Collaborative Network GB. Global Burden of Disease Study 2019 (GBD 2019) Results. Seattle, United States: Institute for Health Metrics and Evaluation (IHME); 2020.” has been changed to the following one, which summarises the main insights of this study.

Murray CJ, Abbafati C, Abbas KM, Abbasi M, Abbasi-Kangevari M, Abd-AllahF et al. Five insights from the global burden of disease study 2019 The Lancet. 2020; p. 1135-1159.

Yours sincerely,

Eduardo Illueca

---

## [Editor Report · Decision Letter 1]

8 Aug 2023

Sequence-Oriented Sensitive Analysis for PM2.5 exposure and risk assessment using Interactive Process Mining

PONE-D-23-03914R1

Dear Dr. Illueca Fernandez,

We’re pleased to inform you that your manuscript has been judged scientifically suitable for publication and will be formally accepted for publication once it meets all outstanding technical requirements.

Kind regards,

Sathishkumar V E

Academic Editor

PLOS ONE
---

## [Editor Report · Acceptance letter]

11 Aug 2023

PONE-D-23-03914R1 

Sequence-Oriented Sensitive Analysis for PM2.5 exposure and risk assessment using Interactive Process Mining 

Dear Dr. Illueca Fernández:

I'm pleased to inform you that your manuscript has been deemed suitable for publication in PLOS ONE. Congratulations! Your manuscript is now with our production department. 

Kind regards, 

on behalf of

Dr. Sathishkumar V E 

Academic Editor

PLOS ONE